# Skin Wound Healing: The Impact of Treatment with Antimicrobial Nanoparticles and Mesenchymal Stem Cells

**DOI:** 10.3390/jox15040119

**Published:** 2025-07-18

**Authors:** Pavel Rossner, Eliska Javorkova, Michal Sima, Zuzana Simova, Barbora Hermankova, Katerina Palacka, Zuzana Novakova, Irena Chvojkova, Tereza Cervena, Kristyna Vrbova, Anezka Vimrova, Jiri Klema, Andrea Rossnerova, Vladimir Holan

**Affiliations:** 1Department of Toxicology and Molecular Epidemiology, Institute of Experimental Medicine CAS, 14220 Prague, Czech Republic; eliska.javorkova@iem.cas.cz (E.J.); michal.sima@iem.cas.cz (M.S.); zuzana.simova@iem.cas.cz (Z.S.); barbora.hermankova@iem.cas.cz (B.H.); katerina.palacka@iem.cas.cz (K.P.); zuzana.novakova@iem.cas.cz (Z.N.); irena.chvojkova@iem.cas.cz (I.C.); tereza.cervena@iem.cas.cz (T.C.); kristyna.vrbova@iem.cas.cz (K.V.); anezka.vimrova@iem.cas.cz (A.V.); andrea.rossnerova@iem.cas.cz (A.R.); vladimir.holan@iem.cas.cz (V.H.); 2Department of Cell Biology, Faculty of Science, Charles University, 12843 Prague, Czech Republic; 3Department of Computer Science, Czech Technical University in Prague, 16000 Prague, Czech Republic; klema@fel.cvut.cz

**Keywords:** skin injury, mesenchymal stem cells, nanoparticles, gene expression, epigenetic regulation, immune response

## Abstract

An investigation into the biological mechanisms initiated in wounded skin following the application of mesenchymal stem cells (MSCs) and nanoparticles (NPs) (Ag, ZnO), either alone or combined, was performed in mice, with the aim of determining the optimal approach to accelerate the healing process. This combined treatment was hypothesized to be beneficial, as it is associated with the production of molecules supporting the healing process and antimicrobial activity. The samples were collected seven days after injury. When compared with untreated wounded animals (controls), the combined (MSCs+NPs) treatment induced the expression of *Sprr2b*, encoding small proline-rich protein 2B, which is involved in keratinocyte differentiation, the response to tissue injury, and inflammation. Pathways associated with keratinocyte differentiation were also affected. Ag NP treatment (alone or combined) modulated DNA methylation changes in genes involved in desmosome organization. The percentage of activated regulatory macrophages at the wound site was increased by MSC-alone and Ag-alone treatments, while the production of nitric oxide, an inflammatory marker, by stimulated macrophages was decreased by both MSC/Ag-alone and MSCs+Ag treatments. Ag induced the expression of *Col1*, encoding collagen-1, at the injury site. The results of the MSC and NP treatment of skin wounds (alone or combined) suggest an induction of processes accelerating the proliferative phase of healing. Thus, MSC-NP interactions are a key factor affecting global mRNA expression changes in the wound.

## 1. Introduction

Skin wound healing is a physiological process that aims to restore the integrity and functionality of damaged tissues. It involves a complex interplay of cellular and molecular events that occur in overlapping stages: hemostasis, inflammation, proliferation, and remodeling [1]. The entire healing process may span a period of up to 100 days [2]. In individual phases, various proteins are involved that are produced by cells such as platelets, neutrophils, macrophages, keratinocytes, endothelial cells, or fibroblasts that support healing [2]. The immune system, which includes both skin-resident and recruited immune cells that serve as a source of cytokines and growth factors, plays an important role in healing [1]. A simplified workflow of wound healing can be described as follows [1]: (1) Hemostasis, the first phase, involving the activation of platelets and the coagulation cascade, mainly serves to stop bleeding. (2) During the inflammatory phase, lasting up to 5 days, cells of the innate and adaptive immune response are recruited. This phase is based on the interaction of damaged cells with Toll-like receptors (TLRs), which result in the activation of signaling pathways leading to the production of inflammatory cytokines (tumor necrosis factor α (TNF-α), interleukin-1 (IL-1)), antimicrobial molecules, and chemokines. (3) The proliferative phase, in which fibroblasts, keratinocytes, and endothelial cells participate, involves re-epithelization of the wound, new blood vessels, and granulation tissue formation. The process is initiated by the release of regulatory factors, including nitric oxide (NO), cytokines, and growth factors (e.g., vascular endothelial growth factor (VEGF-A), platelet derived growth factor (PDGF), basic fibroblast growth factor (bFGF)). (4) Finally, the remodeling phase restores tensile strength as collagen is realigned and scar tissue is formed. Despite this intricate orchestration, wound healing can be impaired due to infection, chronic inflammation, or underlying conditions like diabetes, highlighting the need for advanced therapeutic interventions [3,4].

The application of MSCs has been recognized as an approach that helps and stimulates wound healing (reviewed in [5]). MSCs, derived from sources such as bone marrow, adipose tissue, and umbilical cord, are multipotent stem cells with the ability to regenerate tissues and to modulate immune responses. Clinical trials aimed at the treatment of more than 300 diseases using MSCs are currently underway. So far, studies have shown that such treatments are safe and effective. The route of MSC delivery is critical for the success of the treatment. Currently, local administration directly into the target organ/tissue or systemic delivery by intra-arterial, intravenous, and intraperitoneal injection is used [6]. The optimal delivery method depends on the specific disease, target tissue, and required therapeutic outcome. Among other disorders, skin injuries have also been targeted. MSCs are characterized by low immunogenicity, ease of isolation and expansion, high differentiation potential, and paracrine functions [5]. Specifically, the production of cytokines, chemokines, growth factors, exosomes, and other bioactive components contribute to the therapeutic potential of MSCs. The induction of angiogenesis is another key property of MSCs, as new blood vessels may release cytokines/growth factors (e.g., VEGF, tumor growth factor β (TGF-β), IL-6) that regulate the inflammatory response, cell proliferation, and collagen synthesis, thus further contributing to the healing process. The process of angiogenesis is achieved by both the direct and indirect effects of MSCs on peripheral vascular endothelial cells. Finally, due to their differentiation potential, MSCs contribute to the healing process by the formation of skin tissue cells (keratinocytes, fibroblasts, endothelial cells), as well as by communication with other cell types in the wounded tissue (e.g., macrophages). MSCs have shown the ability to mitigate scar formation by balancing collagen deposition and extracellular matrix (ECM) remodeling. Their immunomodulatory effects, mediated by the secretion of IL-10 and the inhibition of pro-inflammatory cytokines such as TNF-α, further enhance their efficacy in resolving chronic wounds [3]. While MSC-based therapies have shown considerable promise in preclinical and early clinical studies, challenges such as limited cell viability, potential tumorigenicity, and high production costs still remain.

Nanoparticles (NPs) represent another potential approach in wound healing due to their unique physicochemical properties, including a high surface area-to-volume ratio and the ability to be engineered for specific biological interactions [7]. Various NPs, such as silver, zinc oxide, and gold, have demonstrated potent antimicrobial activity, making them effective in preventing wound infections, a major obstacle to efficient healing [4,8]. Silver nanoparticles (Ag NPs), for example, have shown efficacy against multidrug-resistant bacteria while promoting fibroblast proliferation and migration. Zinc oxide NPs (ZnO NPs), in addition to their antibacterial properties, exhibit antioxidative activity, which is critical for mitigating oxidative stress in chronic wounds. NPs can further serve as carriers for drug delivery, enabling the localized and sustained release of therapeutic agents. Polymeric NPs can encapsulate growth factors, antibiotics, or anti-inflammatory drugs, ensuring targeted delivery to the wound site while minimizing systemic side effects. This versatility makes NPs an attractive adjunct to conventional wound care methods [9]. In addition, some NPs may enhance keratinocyte and fibroblast proliferation and suppress the activity of the innate immune response [10], thus supporting wound healing.

The treatment of injured skin with a combination of MSCs and NPs may further enhance the healing properties of these factors alone. These approaches may take advantage of both the induction of the production of molecules supporting healing processes (MSCs and NPs) and antimicrobial activity (if suitable NPs are applied). The combined treatment may result in additive or synergistic effects of both therapeutic options and may thus be superior to the application of individual factors. Metal NPs, in particular, possess some valuable properties that positively influence the repair and growth of wounded tissues. In addition, their small size allows NPs to penetrate cell membranes and to exert their effects intracellularly. However, NPs, especially at higher doses, might be toxic to the target tissue, which may limit their therapeutical applications. This limitation may be reduced by the supporting role of MSCs, while would allow for decreasing NP doses to be used for the treatment. A limited number of reports have recently been published investigating the potential beneficial outputs of such a combined treatment. Gao et al. used a polyvinyl alcohol hydrogel dressing with silver NPs and seeded it with adipose-derived stem cells. The dressing was used to treat skin injury in rats and compared with the dressing without Ag NPs. Although the authors expected an improved healing process due to the presence of Ag, the results for both types of dressings did not differ [8]. Another study combined the use of exosomes derived from bone marrow MSCs with ZnO NPs for the treatment of skin wounds in rats. As a reference, treatment with ZnO NPs alone was used. The treatment was administered daily for 21 days. The authors reported facilitated healing (decreased wound size, increased wound contraction) in exosome-treated animals and suggested that this therapy has potential as a treatment method that modulates angiogenesis, re-epithelization, collagen deposition, and gene expression profiles [11]. Thus, so far, the expected beneficial effects of a combined MSCs+NPs treatment have not been confirmed.

To contribute to the knowledge of the impacts of combined MSCs+NPs administration to injured skin, we used NPs with antimicrobial properties (Ag, ZnO) and MSCs, alone or combined, for wound healing in a murine model. By comprehensive evaluation of the biological processes (including mRNA expression changes, variations in epigenetic settings, and the induction of immune-response-related mechanisms) contributing to these therapies, we focused on elucidating the potential synergistic potential of MSCs and NPs. Addressing these questions will aid in devising strategies to improve wound healing.

## 2. Materials and Methods

### 2.1. Nanoparticle Preparation and Characterization

NPs for the wound healing experiments were prepared and characterized as previously described [12]. Briefly, Ag (NM-300 K; a colloidal dispersion in deionized water supplemented with a stabilizing agent and emulsifiers) and ZnO (NM-110; dry powder) NPs were obtained from the JRC nanomaterials repository. Fresh stock solutions of NPs at a final concentration of 2.56 mg/mL were obtained by sonication and dispersion in sterile water containing 0.05% *w*/*v* BSA. For animal treatment, the stocks were diluted in PBS to the working concentrations as described in Table 1.

### 2.2. Animals and Treatment

In this study, female mice of inbred strain BALB/c were used as donors of the MSCs and the recipients of treatment. Mice of this strain are genetically identical, and no immunological reactivity occurs between donor and recipient (syngeneic conditions). Animals (at the age of 8–15 weeks) were obtained from the Institute of Molecular Genetics of the Czech Academy of Sciences (CAS), Prague, or purchased from Envigo (Indianapolis, IN, USA). The animal experiments were approved by the Local Ethical Committee of the Institute of Experimental Medicine of CAS, Prague (approval code 3058/2021). Female mice were used in order to avoid possible differences in the reactivity between females and males and to avoid the aggressive behavior of male mice toward each other. The ages of the mice correspond to the ages of human adults. In these particular experiments, mice of the same age were regularly used.

Before the treatment, the mice were anesthetized with a xylazine and ketamine mixture (1:1) (Bioveta, Ivanovice, Czech Republic) and secured on an operating pad. After shaving the fur on their backs, a skin lesion of approximately 6 mm in diameter was created. During the procedure, a piece of full-thickness skin was excised using sharp surgical scissors. In all cases, the wound involved the epidermis, dermis, and part of the hypodermis (the average thickness of the excised skin was about 540 µm. The model of an acute skin wound was established, and mice were sacrificed on day 7 after injury, when the wound healing process switches from the inflammatory to the proliferation phase.

Treatment was initiated immediately after injury induction by a single application of the appropriate solution (4 × 10 μL of PBS alone or in combination with 4 × 150,000 MSCs/4 × 125 ng Ag NPs, 4 × 62.5 ng Ag NPs, 4 × 62.5 ng ZnO NPs, 4 × 31.25 ng ZnO NPs). Particular mixtures were prepared immediately before the treatment (without pre-incubation of the MSCs with NPs). The applied volume (40 μL) was divided into four doses (10 μL each) and applied to the wound edge at four opposite points. The treatment combinations are summarized in Table 1. In this study, three or four independent experiments for each method were performed, and all groups in the experiment included at least 2 animals.

The NP concentrations were selected based on previously published data, taking into account the toxicity of NPs when incubated with MSCs [12]. Specifically, the two highest doses not affecting the metabolic activity of MSCs, as assessed by the WST-1 assay, were used for the animal treatments (12.5 and 6.25 μg/mL; 6.25 and 3.12 μg/mL; for Ag NPs and ZnO NPs, respectively). To avoid secondary injury or infection, the wound was covered with a sterile bandage containing a paraffin gauze dressing (Grassolind Neutral; Hartmann, Heidenheim, Germany) and plaster (Appendix A). Mice were divided into groups according to the type of treatment, which was indicated on the bandage using a non-toxic marker.

### 2.3. Mesenchymal Stem Cells: Isolation and Cultivation

Inguinal fat pads were collected from BALB/c mice and digested at 37 °C in a solution of 1 mg/mL collagenase I (Sigma-Aldrich, St. Louis, MO, USA) in Hanks’ Balanced Salt Solution (HBSS, Sigma-Aldrich) with Ca^2+^ and Mg^2+^. After 45 min of digestion, the cell suspension was centrifuged twice at 250× *g* for 8 min in Dulbecco’s Modified Eagle Medium (DMEM; Sigma-Aldrich) containing 10% fetal bovine serum (FBS; Gibco, Grand Island, NY, USA), antibiotics (100 U/mL penicillin, Sigma-Aldrich, 100 µg/mL streptomycin, Sigma-Aldrich), and 10 mM HEPES buffer (Sigma-Aldrich); this is referred to as complete DMEM. The cell suspension was transferred to a 75 cm^2^ tissue culture flask (Techno Plastic Products, Trasadingen, Switzerland) and cultured in complete DMEM.

After 48 h of cultivation, the non-adherent cells were washed off, and the remaining adherent cells were cultured at 37 °C in 5% CO_2_ with regular medium changes and passaging. The cells were used for further experiments at passage 3 (about day 14 of cultivation without cryopreservation).

To assess the effect of NPs on the MSC phenotype, 600,000 cells were cultured in 75 cm^2^ tissue culture flasks for 7 days with selected concentrations of NPs (12.5 and 6.25 µg/mL Ag NPs; 6.25 and 3.12 µg/mL ZnO NPs). In the middle of treatment period, half of the cultivation medium volume was gently removed and an equal volume of fresh complete DMEM was added. The MSC phenotype was then analyzed using flow cytometry (see Section 2.10 for a further description).

### 2.4. Skin Tissue Collection

After seven days, the mice were sacrificed by cervical dislocation, and the tissue from the wound and its surroundings was collected, weighed (to ensure sufficient volume of the lysis buffer in the nucleic acid extraction step), and either evaluated directly or frozen in liquid nitrogen for further analyses. For the control uninjured mice, tissue samples were obtained from the site corresponding to the position of the wound in the injured animals.

### 2.5. Nucleic Acid Extraction for Transcriptomics and the DNA Methylation Assay

DNA, RNA, and miRNA were extracted from skin samples as previously described [13], with some modifications. Briefly, the tissue was homogenized in liquid nitrogen using a mortar and pestle and homogenized with a needle and syringe. For nucleic acid extraction from tissue homogenates, the AllPrep DNA/RNA Mini Kit (Qiagen, Hilden, Germany) was used. The extraction protocol followed the manufacturer’s instructions. The concentration and quality of DNA was assessed using the Nanodrop ND-1000 Spectrophotometer (Thermo Fisher Scientific, Waltham, MA, USA); for RNA, the HS RNA kit with the Qubit 4 fluorometer (Thermo Fisher Scientific, Waltham, MA, USA) and the SS RNA kit with the Fragment Analyzer (Agilent Technologies, Santa Clara, CA, USA) were applied for measuring the concentration and quality, respectively.

### 2.6. mRNA: Library Preparation and Next-Generation Sequencing

To prepare the mRNA libraries, 200 ng of total RNA was used. mRNA obtained using the NEBNext Poly(A) mRNA Magnetic Isolation Module was processed using the NEBNext Ultra II Directional RNA Library Prep kit with Beads and NEBNext Multiplex Oligos for Illumina (all New England Biolabs, Ipswich, MA, USA). Quality control assessment of the libraries was performed using the 1× dsDNA HS kit (Thermo Fisher Scientific) on the Qubit 4 fluorometer and the Fragment Analyzer with the HS NGS Fragment kit (Agilent Technologies). For sequencing, the NVSEQ 6000 SP Reagent Kit v1.5 (100 cycles) and the NovaSeq 6000 system (Illumina, Inc.; San Diego, CA, USA) were used.

### 2.7. miRNA: Library Preparation and Next Generation Sequencing

For generating the miRNA libraries, the total RNA input was 100 ng. The NEXTFLEX Small RNA Sequencing Kit V4 (Revvity, Waltham, MA, USA) was used to prepare the libraries. Quality control assessment of the libraries was performed using the 1× dsDNA HS kit (Thermo Fisher Scientific) on the Qubit 4 fluorometer and the Fragment Analyzer with the HS NGS Fragment kit (Agilent Technologies). Sequencing was performed using NextSeq™ 1000/2000 P2 Reagents (100 cycles) and the NextSeq 1000/2000 system (Illumina).

### 2.8. DNA Methylation

DNA methylation was assessed using Infinium Mouse Methylation BeadChips (Illumina), which allow for the detection of >285,000 CpG markers across the methylome. First, DNA (250 ng/sample) was treated overnight with sodium bisulfite using the Zymo EZ DNA Methylation^TM^ Kit in spin-column format (Zymo Research, Irvine, CA, USA). The concentration of the bisulfite-converted DNA was checked using the Nanodrop ND-1000 Spectrophotometer. Genome-wide DNA methylation was then analyzed using the Infinium HD Methylation Assay. The arrays were scanned by the iScan System (Illumina), and the methylation status at each CpG site was estimated by measuring the intensity of the pair of methylated and unmethylated probes.

### 2.9. The Presence of MSCs in Injured Tissue

To monitor the presence of MSCs in the wound and its surroundings, the cells were labeled prior to their application using the PKH26 Red Fluorescent Cell Linker Kit (Sigma-Aldrich) following the manufacturer’s instructions. To distinguish between vital PKH26-labeled MSCs and phagocytic cells of the recipient that have engulfed cell debris from PKH26-labeled MSCs at the site of injury, staining of the CD45^+^ population was used prior to analysis of the skin samples by flow cytometry. Nevertheless, we cannot absolutely exclude the possibility of passive debris uptake or cell fusion. On day seven of the treatment, the wound, the surrounding tissue and the axillary lymph nodes were collected, and the samples were dissociated mechanically using scissors and then enzymatically digested by collagenase to obtain single-cell suspensions. The samples were further analyzed by flow cytometry as described in Section 2.10.

### 2.10. Flow Cytometry

Flow cytometry was used for several analyses, using different antibodies as reported in Table 2. For skin lesion samples, the single-cell suspensions were prepared by tissue digestion in 1 mg/mL collagenase II (Sigma-Aldrich) in HBSS at 37 °C for 90 min. Before the analysis, the samples were washed in PBS (Sigma-Aldrich) and incubated for 30 min at 4 °C with anti-mouse monoclonal antibodies (all purchased from BioLegend, San Diego, CA, USA) conjugated to allophycocyanin (APC), phycoerythrin (PE), or fluorescein isothiocyanate (FITC). For MSC characterization, cells stained with PE-labeled rat IgG2a (clone RTK2758), APC-labeled rat IgG2b (clone RTK4530), or FITC-labeled rat IgG2b (clone RTK4530) were used as negative controls. To identify dead cells, Hoechst 33258 fluorescent dye (Invitrogen, Carlsbad, CA, USA) was added to the suspension 10 min before measurement. Samples were analyzed using an LSRII flow cytometer (BD Biosciences, Franklin Lakes, NJ, USA), and the resulting data were processed with FlowJo software (version 9) (BD Biosciences).

### 2.11. Enzyme-Linked Immunosorbent Assay (ELISA) and Detection of NO Production

To detect the production of pro-inflammatory cytokines and nitric oxide (NO), the excised skin lesions were cut into small pieces and cultured for 48 h in 700 µL of RPMI 1640 medium (Sigma-Aldrich) containing 10% fetal bovine serum (FBS, Gibco), antibiotics (100 U/mL penicillin, 100 µg/mL streptomycin), and 10 mM HEPES buffer; this is referred to as complete RPMI. Skin lesions were cultured in the presence of 5 µg/mL lipopolysaccharide (LPS) (Sigma-Aldrich) and 10 ng/mL IFN-γ (PeproTech, Rocky Hill, NJ, USA). The culture supernatants were analyzed using DuoSet ELISA kits (R&D Systems, Minneapolis, MN, USA) following the manufacturer’s instructions. The concentration of NO in the supernatant was measured using the Griess reaction by adding a 1:1 mixture of 1% sulfanilamide (Sigma-Aldrich) and 0.3% N-1-naphthylethylenediamine dihydrochloride (Sigma-Aldrich) (both in 3% H_3_PO_4_) (Sigma-Aldrich). The resulting reaction was quantified using the Sunrise Remote ELISA Reader (Gröding, Austria).

### 2.12. Real-Time PCR

Skin samples were collected in 1000 µL of TRI Reagent (Molecular Research Center, Cincinnati, OH, USA), frozen at −80 °C, and homogenized after thawing. Total RNA was isolated according to the manufacturer’s instructions. Reverse transcription was performed using deoxyribonuclease I (Promega, Madison, WI, USA) and buffer (Promega). In the subsequent step, M-MLV reverse transcriptase with buffer (Promega), RNAsin (Promega), dNTPs (Promega), and random primers (Promega) were used to synthesize the cDNA. The total reaction volume was 25 µL. Quantitative RT-PCR was conducted using the SYBR Green Master Mix (Applied Biosystems, Foster City, CA, USA) on a StepOne-Plus Real-Time PCR System (Applied Biosystems) with the following parameters: denaturation at 95 °C (3 min), followed by 40 cycles at 95 °C (20 s), annealing at 60 °C (30 s), and elongation at 72 °C (30 s). The data were collected following elongation at 80 °C for 5 s at each cycle. The obtained data were analyzed with StepOne software version 2.3 (Applied Biosystems). We analyzed the amplification curves and melt curves to verify the specificity of the primers and the purity of the PCR products. Relative gene expression was calculated using the relative quantification model compared with glyceraldehyde-3-phosphate dehydrogenase (*Gapdh*). The stability of the reference gene was investigated using the geNorm application. The primers (Generi Biotech, Hradec Kralove, Czech Republic) used for amplification are listed in Table 3.

### 2.13. Data Analysis

For analysis of the transcriptomics data, we utilized the nf-core pipelines for standardized and reproducible RNA-seq data analysis [14]. These pipelines integrate various bioinformatics tools for quality control, alignment, and quantification, ensuring a consistent workflow with robust results. Specifically, we employed the nf-core/rnaseq pipeline version 3.2 with the GRCm38 reference genome for mRNA FASTQ processing and the nf-core/smrnaseq pipeline version 2.2.4 with the same reference genome, and the Nextflex protocol was used for miRNA quantification.

For DNA methylation data analysis, a previously published approach was used [15]. Raw microarray data were downloaded as idat files, imported into the R environment and processed with the sesame package. Data were normalized using the quantile method. Beta values for the determination of the level of methylation as the ratio of the fluorescent signals from the methylated vs. unmethylated sites were calculated using the sesame package. Preprocessing analyses were performed to study the distribution of beta values and the variation in methylation across all samples.

For other experimental data, GraphPad Prism version 9.0.0 (GraphPad Software, Boston, MA, USA) was used. The results are expressed as the mean+SEM. The statistical significance of differences between individual groups was calculated using the Student’s *t*-test.

## 3. Results

### 3.1. Whole Genome mRNA and miRNA Expression Induced by Individual Treatment Scenarios

Skin injury was the most significant factor affecting the whole-genome mRNA expression profiles: a total of 2006 mRNAs were downregulated and 2216 were upregulated as a result of wound infliction. Further changes in mRNA numbers resulting from the individual treatment scenarios were rather modest (Table 4; for a complete list of deregulated mRNAs, see Appendix A; FASTQ files are stored in the public repository at: Skin Wound Healing_mRNA expression), resulting in a maximum of 216 deregulated mRNAs (from the application of Ag-L alone). Interestingly, treatment with ZnO alone (regardless of the dose) had no impact on the number of deregulated mRNAs. The combined MSCs+ZnO treatment affected about twice as many mRNAs than the combined MSCs+Ag treatment. The application of MSCs alone affected the expression of seven mRNAs, of which three were unique to this type of treatment (*Acot11*, *Lair1*, *Pyhin1*) (Figure 1). It should be further noted that despite the differences in the number of deregulated genes between samples treated with Ag and ZnO NP, there were common genes affected by both types of NPs (Figure 2). The genes common to at least three treatment conditions include *Aldh6A1* (MSCs+Ag-L, MSCs+ZnO-L, and MSCs+ZnO-H) and *Sprr2b* and *Slc9b2* (MSCs+Ag-L, MSCs+Ag-H, MSCs+ZnO-L, and MSCs+ZnO-H). The expression of *Sprr2a3*, *Alox8*, *Chil4*, *Rnase2a*, *Ocstamp*, and *Atp10b* was commonly affected by the MSCs+Ag-H, MSCs+Ag-L, and MSCs+ZnO-L treatments.

The pathway analysis identified several common deregulated pathways: for MSCs+Ag-H and MSCs+ZnO-H, the “neutrophil degranulation” and “innate immune system” pathways were detected, while for MSCs+Ag-L and MSCs+ZnO-L, pathways involved in “formation of the cornified envelope” and “keratinization” were found (Table 5). In addition, for MSCs+ZnO-L, “fatty acid biosynthesis” and “fatty acid metabolism” pathways were affected. The treatment with Ag-L alone affected pathways involved in “neutrophil degranulation”, the “innate immune system”, and the “immune system”.

In contrast to mRNAs, the number of miRNAs deregulated as a result of skin injury and subsequent treatment was very low. The skin injury itself was the most significant factor, causing the deregulation of a total of 264 miRNAs (of which 123 were downregulated and 141 were upregulated). The treatment (regardless of its type) had an impact on these very low numbers of miRNAs, suggesting that the biological significance of this epigenetic regulatory mechanism was minimal (Table 6; for a complete list of deregulated miRNAs, see Appendix A; FASTQ files are stored in the public repository at: Skin Wound Healing_miRNA expression). For that reason, no further investigation involving miRNA deregulation was conducted.

### 3.2. DNA Methylation Changes

Similarly to the mRNA and miRNA expression changes, differential CpG site methylation was mainly affected by the wound infliction itself: 104,534 CpG sites in 12,701 individual genes were identified to be significantly modified seven days after skin injury in the untreated control. Of interest, MSC treatment alone had no impact on CpG methylation when compared with untreated injured skin; this was also true for the application of ZnO (both doses), Ag-H, and the combination of MSCs+Ag-L. Overall, using Ag for wound treatment impacted a greater number of CpG sites than ZnO application (a total of 1031 vs. 405 unique CpG sites; an overview of the number of all the affected sites is reported in Table 7). Despite these differences, common genes in which differentially methylated CpG sites are located were found (Figure 3; Appendix A; raw data are stored in the public repository at: Skin Wound Healing_DNA methylation): a combined treatment with MSCs+ZnO-L affected several genes identical to those from treatments in which Ag NPs were used. In addition, some of the processes, in which genes with differentially methylated CpG sites induced by Ag vs. ZnO NPs were involved, overlapped. They included “platelet activation/signaling and aggregation”, “epithelial cell development” and “positive regulation of cell migration” (Appendix A). Regarding the application of Ag NPs, three common processes that included genes with significantly differentially methylated CpG sites were found for the MSCs+Ag-H and Ag-L treatments: “desmosome organization”, “negative regulation of cell population proliferation” and “positive regulation of phosphatidylinositol 3-kinase/protein kinase B signal transduction” (Appendix A). A potential association between CpG site methylation and expression changes in the corresponding mRNAs was investigated, but no clear link could be identified.

### 3.3. Phenotypic Characterization of MSCs: The Impacts of NPs

The in vitro cultivation of MSCs with NPs and the subsequent flow cytometry analysis revealed no significant effects on the expression of surface molecules, including negative leukocyte markers (CD45, CD11b, and CD31) and positive MSC-associated markers (CD105, CD90, CD44, and CD106) (Figure 4A). No differences for each of the tested concentrations were found.

### 3.4. The Presence of MSCs at the Injury Site and Surrounding Tissue

After the seven days of skin injury treatment, approximately 1% of the applied fluorescently labeled MSCs was detected at the wound site. The number of MSCs presented in the surrounding tissue was about 30× lower, and in the axillary lymph nodes, an insignificant number of MSCs was detected (Figure 4B and Appendix A). This indicates that the migration of MSCs from the site of injury was minimal. No differences between individual treatment groups were observed. In addition, no effect of NPs on the phagocytosis of MSCs by leukocytes present at the wound site was detected.

### 3.5. The Analysis of Leukocyte Populations Infiltrating the Injury

No significant changes in the proportion of T-cells, B-cells, granulocytes (Appendix A), or macrophages (F4/80^+^ cells) (Figure 5A and Appendix A) were detected in samples from the wound site seven days after the application of MSCs and/or NPs. However, the proportion of activated regulatory macrophages (CD80^+^CD206^+^ cells) among the macrophages detected in the wound was significantly elevated in the samples treated with MSCs alone and Ag-L alone (Figure 5A). Overall, this observation suggests positive effects of separate MSC and Ag NP skin injury treatment, as manifested by the increased percentage of activated regulatory macrophages.

Further, we focused on the potential effect of NPs on MSCs. The treatment combining MSCs and Ag-L was shown to reduce the macrophage infiltration of the wound more effectively than MSCs alone, while a treatment combining MSCs and Ag NPs seemed to act negatively and abrogated the effect of separate MSCs on the induction of regulatory macrophages.

### 3.6. The Effect of MSC and NP Treatment on Macrophages in Skin Wounds

Tissue samples containing macrophages infiltrating the skin lesion were stimulated with LPS and INF-γ, and the production of NO and selected inflammatory factors (TNF-α and IL-6) was measured by the Griess reaction and ELISA, respectively. A decrease in the analyzed parameters following various combinations of treatment was detected. Specifically, a significantly lower production of NO was detected after the administration of MSCs alone, a combination of MSCs+Ag-L, and Ag-H alone, suggesting a significant impact of the treatment on activated macrophages. In addition, Ag-H alone decreased TNF-α and IL-6 production (Figure 5B).

The analysis of the effect of NPs on MSCs showed that MSCs alone significantly reduced the production of NO by activated macrophages, while the combination of MSCs with a higher dose of Ag NPs abrogated this effect.

### 3.7. Analysis of the Effect of MSCs and NPs on the Processes Occurring During the Early Phase of Wound Healing

The RT-PCR analysis of selected immunomodulation- and regeneration-related molecules did not, for the most part, show an effect of Ag NPs on the processes induced by MSCs. As shown in Figure 6, the expression of galectin-3, chemokine CCL-2, matrix metalloproteinase-2 (MMP-2), and hepatocyte growth factor (HGF) mRNAs, which reflect the processes related to regeneration and late inflammation at the site of injury, was increased in the wounded, untreated skin. The expression of VEGF mRNA, indicating a pro-angiogenic environment within the injured skin, was not significantly changed in any of the samples. Finally, the expression of collagen1 (COL1) mRNA was induced in skin samples treated with Ag-L alone, further suggesting an induction of regeneration processes.

### 3.8. Wound Healing—Macroscopic Observations

After one week, lesions in both the control and treated animals had begun to heal spontaneously, suggesting that the injury was relatively mild due to the absence of pathogens and the lack of severely damaging agents (e.g., burns or chemical exposure). A detailed analysis of the molecular and cellular mechanisms involved in wound healing on day 7 after the treatment with MSCs and/or NPs was the major aim of this study; thus, no information on other time points is provided. No visible differences between individual treatment scenarios were noted (Appendix A).

## 4. Discussion

Our study aimed to compare the effect of MSCs and a combination of MSCs+NPs with antimicrobial properties on skin wound healing in mice. The experiments consisted of two parts: (1) Whole genome mRNA expression and epigenetic regulation (miRNA expression, DNA methylation) changes were evaluated for Ag and ZnO NPs. (2) Based on the results, Ag NPs were used for further analyses of immune-response- and regeneration-related molecules. NPs for the skin injury experiments were selected based on a previous study in which the effects of metal NPs on the characteristics and functional properties of MSCs were tested in vitro [12]. For our animal experiments, we selected the highest and the second-to-highest dose of the respective NP that was not toxic in the in vitro tests. ZnO administered alone had generally minimal biological effects when compared with Ag NPs, which might be related to the non-optimal dose selection or, rather, to the different physico-chemical properties of the tested NPs, as discussed further. To the best of our knowledge, this is the only study that has used such a comprehensive scenario to assess the mechanisms induced by a combination of MSCs+NPs in the healing of skin injury.

### 4.1. mRNA Expression and Epigenetic Modulation

As expected, the skin injury itself, without any treatment, was the factor with the most pronounced effects on mRNA expression. This observation is in agreement with substantial biological changes known to be initiated by wound infliction [1]. In our study, the samples were collected seven days after injury, i.e., in the phases of inflammation and proliferation. As a result, neutrophils, macrophages, fibroblasts, and endothelial cells were expected to be present in the affected tissue, and collagen was expected to be produced [2]. Considering the biological properties of MSCs, it is rather unexpected that the administration of MSCs alone affected the expression of only seven mRNAs when compared with wounded, untreated animals. Of these molecules, three were unique to MSC administration: *Acot11* (encoding acyl-CoA thioesterase 11, a protein involved in fatty acids metabolism; its role in wound healing might be related to its ability to affect the invasive potential of the cells), *Lair1* (encoding leukocyte-associated immunoglobulin-like receptor 1, a molecule expressed on immune cells that is crucial for maintaining immune homeostasis and for promoting tolerogenic immune response that helps to suppress excessive inflammation and promote tissue repair and regeneration), and *Pyhin1* (encoding pyrin and HIN domain family member 1, a molecule responsible for DNA recognition and regulation of the inflammatory response; its role in wound healing, however, is not clear yet). Adding NPs to the treatment scenarios clearly showed the importance of MSC-NP interactions for achieving a substantial molecular response; while treatments with NPs alone had mostly no effect on the number of deregulated mRNAs (Ag-L being a notable exception), simultaneous administration of MSCs and NPs affected the expression of up to 171 mRNAs (MSCs+ZnO-L). Genes discussed further were ones that were commonly deregulated by at least three treatment conditions. *Sprr2b* and *Slc9b2* were both commonly induced by all the tested MSC+NP combinations (MSCs+Ag-L, MSCs+Ag-H, MSCs+ZnO-L, and MSCs+ZnO-H). *Sprr2b*, encoding small proline-rich protein 2B, is involved in keratinocyte differentiation in response to tissue injury and inflammation. These roles were also recently reported in human keratinocytes [16]. The encoded protein is crucial for maintaining epidermal homeostasis and for providing protection against bacterial infection. Our observation shows the involvement of the tested combinations of MSCs+NPs in the induction of healing processes. It should be noted that MSCs or NPs applied separately had no effect on the expression of this gene, indicating the importance of the combined treatment on the wound healing. *Slc9b2* encodes a member of the solute carrier (SLC) family of ion transporters. This protein contributes to cellular pH regulation and may influence cellular signaling pathways, both being generic mechanisms not directly linked to wound healing.

Combined treatments with MSCs+Ag-H, MSCs+Ag-L, and MSCs+ZnO-L all induced the expression of *Sprr2a3*, *Alox8*, *Chil4*, *Rnase2a*, *Ocstamp*, *Atp10b*. Apart from *Ocstamp*, which encodes a transmembrane protein involved in osteoclast differentiation, thus making its role in the healing of the wounded skin unclear, functions of all these genes are related to the expected processes at the injury site: *Sprr2a3* encodes small proline-rich protein 2A3 with a function similar to that of Sprr2b; *Alox8* encodes arachidonate 8-lipoxygenase, which is involved in the production of mediators that play a role in inflammation, immune response, and tissue repair; *Chil4* encodes chitinase-like protein 4, which is expressed during inflammation and tissue remodeling; *Rnase2a* encodes ribonuclease A2, an enzyme important for antimicrobial activity and which also modulates the immune response and inflammation. *Atp10b,* which encodes ATPase phospholipid transporting 10B, the enzyme essential for membrane function and stability, was the only gene whose expression was downregulated by the treatment. Thus, its biological role in our experimental setting is not clear.

*Aldh6A1* (induced in common by MSCs+Ag-L, MSCs+ZnO-L, and MSCs+ZnO-H treatment) encodes the aldehyde dehydrogenase 6 family member A1 that plays a role in the metabolism of fatty acids and some amino acids. Similarly to *Atp10b*, its expression was downregulated. As this enzyme contributes to the energy production of cells, which might be expected during the healing process, the reason for the common downregulation of the gene is not clear. However, the functions in which the enzymes are involved are rather generic and not specific to the mechanisms linked with MSC activities in damaged tissues. Thus, the biological significance of this observation is not clear, and it might be a spurious finding.

The pathway analysis showed similarities in the impact of MSCs+NPs and NPs alone on the processes involved in wound healing. “Neutrophil degranulation” along with “innate immune system” were the pathways affected by Ag-L and MSCs+Ag-H/MSCs+ZnO-H. Neutrophil degranulation is an antimicrobial mechanism in neutrophils that is based on the release of proteases from intracellular vesicles [17]. The treatment with MSCs+Ag-L/MSCs+ZnO-L modulated the “formation of the cornified envelope” and “keratinization” pathways. Both processes refer to the formation of filaments that help to maintain the mechanical stability of cells and the epithelial tissue during healing [18]. Overall, these results suggest comparable effects of the tested combinations of MSCs+NPs that depend on the dose of NPs: for higher doses, an inflammatory response is preferred (the earlier phase of the healing process), while the lower NP dose is, instead, linked with collagen formation, i.e., phases of proliferation and remodeling. For MSCs+ZnO-L, deregulation of the “fatty acid metabolism” process was further detected. This observation is in agreement with the metabolic reprogramming initiated in skin wounds that also includes changes in fatty acid synthesis. Lipids are the source of energy and support the angiogenic activity of MSCs [19].

We further aimed to address the question of how epigenetic regulation is involved in the response to skin injury and the subsequent healing processes. Our data show that the role of miRNA is non-significant. In contrast, DNA methylation was affected by both types of NPs, although the response differs. While ZnO NPs need to be administered with MSCs to induce changes in the CpG site methylation profiles, for Ag NPs, both treatment scenarios (with and without simultaneous MSC application) affect these types of epigenetic changes, although the response differed for Ag-H and Ag-L doses. While for Ag-H, a combination with MSCs was needed to induce DNA methylation changes, Ag-L was effective only when administered alone. The explanation of such a result is not clear; however, it can be speculated that interactions between NPs and MSCs that induce biological response require a certain “optimal” ratio between the numbers of both factors.

Interestingly, the presence of Ag was linked with the modulation of the “desmosome organization” pathway. Desmosomes are intracellular junctions that participate in the formation of the cytoskeletal network by mechanical integration of the adjacent cells [20]. Another affected pathway, “phosphatidylinositol 3-kinase/protein kinase B signal transduction (PI3K/AKT)”, is also known to participate in the healing process, particularly in cell proliferation and angiogenesis [21,22]. Although the impact of MSCs/NPs on DNA methylation changes in the healing wound is clear, it should be noted that the analyses indicate changes in epigenetic regulation but cannot specifically identify the modifications leading to the upregulation and downregulation of gene expression.

Overall, the genomic and epigenetic experiments showed the importance of the presence of NPs in the healing process in which MSCs were included. While there are common molecular impacts for both types of NPs, we selected Ag NPs for further investigation. The reason for this is the wider use of this type of NPs in clinical practice, as well as the unique impacts of Ag NPs on the epigenetic processes involved in cytoskeleton-remodeling PI3K/AKT signaling. In addition, our in vitro data suggest a more pronounced effect of Ag NPs on MSC differentiation and growth factor production [12].

### 4.2. The Impacts of Treatment on Immunomodulatory and Regeneration-Related Molecules

The following experiments were designed to make sure that the therapy with MSCs/NPs induces the cellular processes that regulate the immune response and accelerate healing. First, we checked the in vitro impact of NPs (both Ag and ZnO) on the phenotypic characteristics of MSCs and found no significant effects for either type at any tested dose. This was an important observation, confirming that the biological properties of MSCs would be preserved in the wound even in the presence of NPs. Although such experiments have also been conducted for other NP types (e.g., [23]), to the best of our knowledge, our report is the only one focusing on Ag and ZnO NPs.

The presence of MSCs at the injury site is another important prerequisite for effective wound treatment. We confirmed that at the time of tissue collection (i.e., seven days after application), there was approximately 1% of the initially applied MSCs present in the wound; very few had migrated to the surrounding tissue and virtually none were detectable in the axillary lymph nodes. A comparable result was obtained for the samples treated with Ag NPs, confirming no negative effects of NPs on the presence of MSCs at the injury site.

Macrophages play a key role in the healing process [24]. First, during the inflammatory phase, they exert pro-inflammatory functions, including the production of cytokines and growth factors. Subsequently, in the proliferative phase of healing, they stimulate fibroblasts, keratinocytes, and endothelial cells to induce extracellular matrix formation and angiogenesis. We thus checked for the presence of macrophages (F4/80^+^ cells) at the injury site after individual treatment combinations. No significant impacts on the percentage of macrophages for MSCs, Ag, or MSCs+Ag treatment was detected when compared with injured, untreated tissue. This observation could be related to the fact that on day seven, the healing process should have progressed to the proliferative phase, when the activity of macrophages decreases. We further checked for the percentage of activated regulatory macrophages (CD80^+^CD206^+^ cells) among all macrophages. This subset of macrophages is anti-inflammatory and is responsible for the attenuation of the immune response, which is expected in the proliferative phase of healing. The treatment with MSCs alone and Ag-L alone significantly increased the percentage of these macrophages, suggesting their beneficial effects on the healing process.

Further experiments were performed to identify the potential reduction in the inflammatory response mediated by macrophages in the wound after the application of various treatments. The production of inflammatory markers (NO, TNF-α, and IL-6) was investigated in the skin lesion samples stimulated by IFN-γ and LPS [25]. The most pronounced response was found for NO production, which is considered the gold standard when measuring the induction of inflammation [26]. Namely, applications of MSCs alone, Ag-H alone, and a combination of MSCs+Ag-L significantly reduced the production of NO, suggesting anti-inflammatory effects of these treatment approaches. The production of TNF-α and IL-6, cytokines produced during the inflammation phase [27], was only significantly decreased by Ag-H administration to the wound. Overall, the results indicate that at the time of sample collection (day seven), treatment is associated with a reduction in the inflammatory response, thus supporting the presence of a proliferative phase in the healing process.

To confirm the induction of proliferative/regenerative processes in the wound at the time of sample collection, we analyzed the mRNA expression of selected genes, including those encoding galectin 3 (implicated in the regulation of inflammation, angiogenesis, and re-epithelization [28]); CCL-2 (an anti-inflammatory molecule involved in wound healing [29]; HGF (a molecule supporting migration of keratinocytes [30]); MMP-2 (a protease responsible for the regulation of inflammation and degradation of the extracellular matrix [31]), collagen-1 (a key structural molecule of the extracellular matrix [32]); and VEGF (a factor that plays a role in angiogenesis and wound closure [33]). The data mostly did not show any significant response; however, the expression of *Col1*, encoding collagen-1, was significantly induced by Ag-L treatment, indicating support for the healing process by the application of Ag NPs.

## 5. Conclusions

In our study, we focused on an investigation of the molecular and cellular processes associated with skin wound treatment using MSCs and NPs (Ag/ZnO), alone or in combination. To the best of our knowledge, this is the only report in which such a complex experimental approach has been used. At the level of global mRNA expression, we identified changes in the expression of the *Sprr2b* gene, which is involved in keratinocyte differentiation, that were induced by a combined MSC+NP (both Ag and ZnO) treatment. Similar effects were observed in the pathway analysis, in which deregulation of keratinization processes were identified after the application of MSCs and a lower dose of both types of NPs. Interestingly, epigenetic changes involving desmosome organization were induced by Ag NPs (alone or in combination with MSCs) but not by ZnO NPs. Overall, the genomics data suggest the importance of NP application, particularly Ag NPs, in combination with MSCs or alone, on the processes linked with the proliferative phase of wound healing. The analyses of the expression of immunomodulatory molecules also suggested the induction of processes associated with the proliferative/regeneration phase of healing. However, the role of individual treatment scenarios was not consistent; rather, the data suggest a beneficial impact of separate Ag and/or MSC application. We speculate that the more pronounced biological effects of Ag NPs than those of ZnO NPs might be related to different physico-chemical properties of both NP types. However, the confirmation of this statement was beyond the scope of our study. In summary, our results clearly show the induction of processes that may accelerate the proliferative phase of wound healing induced by MSCs and NPs. However, follow-up studies are needed to clarify the role of a combined MSCs+Ag treatment.

## Figures and Tables

**Figure 1 jox-15-00119-f001:**
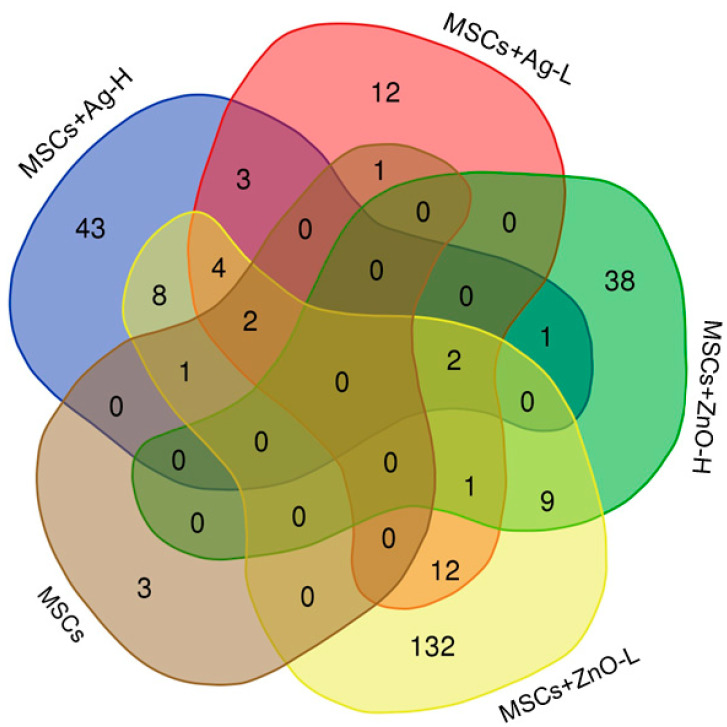
A Venn diagram of the number of unique and common deregulated mRNAs resulting from skin injury treatment with MSCs and a combination of MSCs+NPs (using the wounded but untreated skin group as the reference).

**Figure 2 jox-15-00119-f002:**
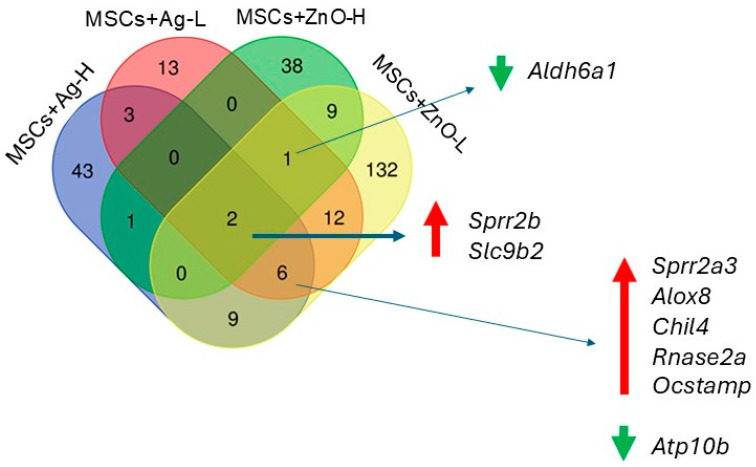
A Venn diagram of the number of unique and common deregulated mRNAs resulting from skin injury treatment with combinations of MSCs+Ag/ZnO NPs at various doses of NPs (using the wounded but untreated skin group as the reference). Green arrows denote downregulation while red arrows indicate upregulation of mRNA expression.

**Figure 3 jox-15-00119-f003:**
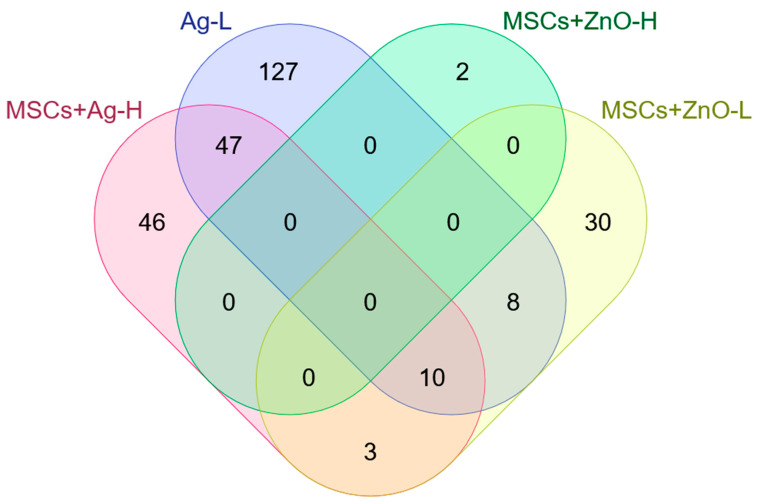
A Venn diagram of the number of unique and common genes in which significantly differentially methylated CpG sites were identified.

**Figure 4 jox-15-00119-f004:**
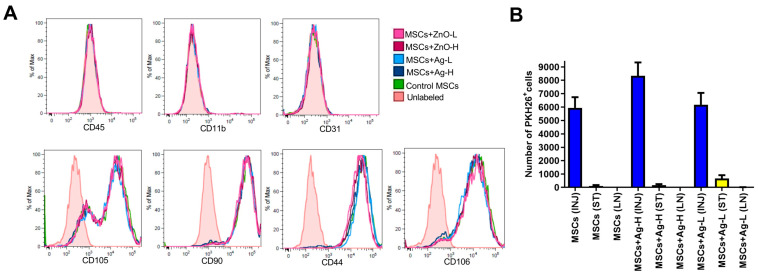
(**A**) Flow cytometry analysis of MSC surface markers in samples treated with selected concentrations of Ag and ZnO NPs. MSCs were cultured for seven days with NPs and then analyzed by flow cytometry. Representative histograms from 3 independent measurements are shown. The pink-tinted histogram represents the control unlabeled MSCs. (**B**) The persistence of MSCs at the injury site, in the surrounding tissue, and in the draining lymph nodes. Fluorescently labeled MSCs were detected by flow cytometry analysis as the number of CD45^−^PKH26^+^ cells in a single-cell suspension prepared from the injury site (INJ), surrounding tissue (ST), or axillary lymph nodes (LN). Each bar represents the mean+SEM from 4 independent experiments.

**Figure 5 jox-15-00119-f005:**
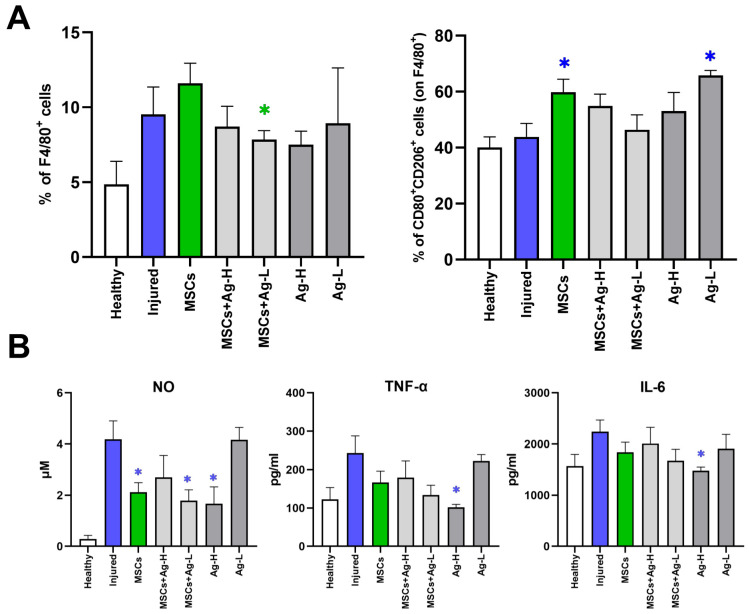
(**A**) The percentage of macrophages (F4/80^+^ cells) and activated regulatory macrophages (expressed as the percentage of CD80^+^CD206^+^ cells among the F4/80^+^ population) at the injury site was assessed by flow cytometry. Each bar represents the mean+SEM from 3 independent experiments. Values with an asterisk are significantly different (* *p* < 0.05) from the control value (green asterisk: the group treated with MSCs alone was used as a reference; blue asterisk: the injured group was used as a reference). (**B**) Analysis of the production of selected inflammatory molecules secreted by activated macrophages. The production of NO, TNF-α, and IL-6 in skin wound samples stimulated with IFN-γ and LPS was assessed by the Griess reaction and ELISA. Each bar represents the mean+SEM from 3 independent experiments. Values with an asterisk are significantly different (* *p* < 0.05) from the control value (green asterisk: the group treated with MSCs alone was used as a reference; blue asterisk: the injured group was used as a reference).

**Figure 6 jox-15-00119-f006:**
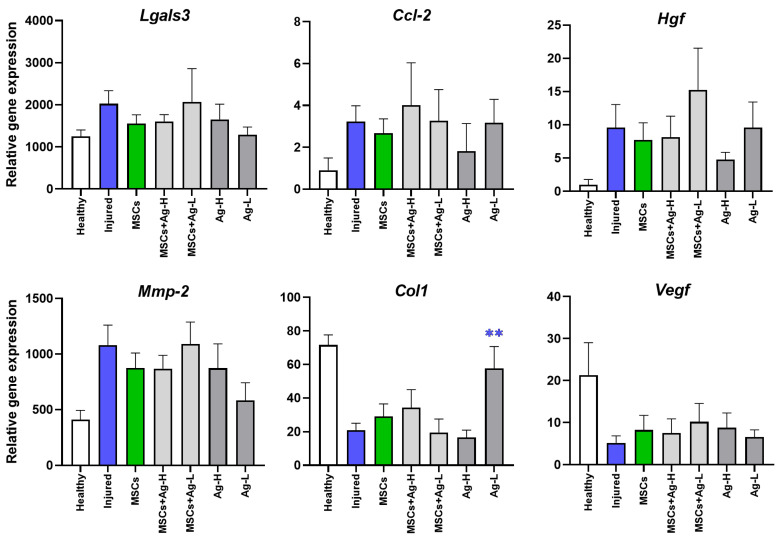
The expression of selected immunomodulatory and regenerative molecules as assessed by RT-PCR in samples collected from the sites of injury. Changes in the expression of galectin-3 (*Lgals3*), chemokine CCL-2 (*Ccl-2*), hepatocyte growth factor (*Hgf*), matrix metalloproteinase-2 (*Mmp-2*), collagen1 (*Col1*), and vascular endothelial growth factor (*Vegf*) were analyzed in skin wounds 7 days after the application of different types of treatment by RT-PCR. Each bar represents the mean+SEM from 4 independent experiments. A significant difference (** *p* < 0.01) when compared with the control value (injured group) was observed for *Col1* expression only; all other gene expression changes were not statistically significant.

**Table 1 jox-15-00119-t001:** Treatment of wounded skin with MSCs and/or NPs: an overview of experimental scenarios.

Sample Type	Treatment Combination
Healthy	None
Injured only	Wound inflicted; 4 × 10 μL of PBS
Injured, MSCs	Wound inflicted; 600,000 MSCs in PBS
Injured, MSCs+Ag-H	Wound inflicted; 600,000 MSCs+Ag NPs (12.5 μg/mL) in PBS
Injured, MSCs+Ag-L	Wound inflicted; 600,000 MSCs+Ag NPs (6.25 μg/mL) in PBS
Injured, Ag-H	Wound inflicted; Ag NPs (12.5 μg/mL) in PBS
Injured, Ag-L	Wound inflicted; Ag NPs (6.25 μg/mL) in PBS
Injured, MSCs+ZnO-H	Wound inflicted; 600,000 MSCs+ZnO NPs (6.25 μg/mL) in PBS
Injured, MSCs+ZnO-L	Wound inflicted; 600,000 MSCs+ZnO NPs (3.12 μg/mL) in PBS
Injured, ZnO-H	Wound inflicted; ZnO NPs (6.25 μg/mL) in PBS
Injured, ZnO-L	Wound inflicted; ZnO NPs (3.12 μg/mL) in PBS

**Table 2 jox-15-00119-t002:** An overview of the antibodies and endpoints for which flow cytometry was used.

Antibody	Fluorochrome	Clone	Endpoint
anti-CD3	APC	17A2	Infiltration of skin wound
anti-CD11b	APC	M1/70	MSC characterization/Infiltration of skin wound
anti-CD19	FITC	6D5	Infiltration of skin wound
anti-CD31	PE	MEC13.3	MSC characterization
anti-CD45	APC	30-F11	Migration of MSCs
anti-CD45	FITC	30-F11	MSC characterization
anti-CD80	APC	16-10A1	Infiltration of skin wound
anti-CD90	FITC	30-H12	MSC characterization
anti-CD44	APC	IM7	MSC characterization
anti-CD105	PE	MJ7/18	MSC characterization
anti-CD106	PE	429	MSC characterization
anti-CD206	FITC	C068C2	Infiltration of skin wound
anti-F4/80	PE	BM8	Infiltration of skin wound
anti-Gr-1	FITC	RB6-8C5	Infiltration of skin wound

**Table 3 jox-15-00119-t003:** Sequences of primers used for RT-PCR analyses.

Gene	Forward Primer	Reverse Primer
*Col-1*	CTCCGGCTCCTGCTCCTCTT	ACTCGCCCTCCCGTCTTTGG
*Gapdh*	AGAACATCATCCCTGCATCC	ACATTGGGGGTAGGAACAC
*Vegf*	AAAAACGAAAGCGCAAGAAA	TTTCTCCGCTCTGAACAAGG
*Hgf*	CACCCCTTGGGAGTATTGTG	GGGACATCAGTCTCATTCACAG
*Lgals3*	TGCGTTGGGTTTCACTGTGCC	GGTGCCCTATGACCTGCCCT
*Ccl-2*	CGGCGAGATCAGAACCTACAAC	GGCACTGTCACACTGGTCACTC
*Mmp-2*	CCCTCAAGAAGATGCAGAAGTTC	ATCTTGGCTTCCGCATGGT

**Table 4 jox-15-00119-t004:** The number of mRNAs deregulated as a result of skin injury, seven days after wound infliction. A comparison was made with the wounded but untreated skin group as a reference.

Group	Downregulated	Upregulated	Total Deregulated
MSCs	0	7	7
MSCs+Ag-H	21	43	64
MSCs+Ag-L	18	19	37
Ag-H	0	0	0
Ag-L	68	148	216
MSCs+ZnO-H	39	12	51
MSCs+ZnO-L	123	48	171
ZnO-H	0	0	0
ZnO-L	0	0	0

**Table 5 jox-15-00119-t005:** Identification of the pathways deregulated by skin injury treatment with various combinations of MSCs+Ag/ZnO NP. The changes are related to the untreated wounded skin group as a reference.

Group	Total Deregulated mRNAs	Affected Pathways
MSCs+Ag-H	64	Neutrophil degranulation; Innate immune system
MSCs+Ag-L	37	Formation of the cornified envelope; Keratinization
Ag-L	216	Neutrophil degranulation; Innate immune system; Immune system
MSCs+ZnO-H	51	Neutrophil degranulation; Innate immune system
MSCs+ZnO-L	171	Formation of the cornified envelope; Keratinization; Fatty acid biosynthesis; Fatty acid metabolism

**Table 6 jox-15-00119-t006:** Number of miRNAs deregulated as a result of skin injury, seven days after wound infliction. A comparison was made with the wounded but untreated skin group as a reference.

Group	Downregulated	Upregulated	Total Deregulated
MSCs	1	0	1
MSCs+Ag-H	0	0	0
MSCs+Ag-L	0	0	0
Ag-H	1	0	1
Ag-L	0	1	1
MSCs+ZnO-H	1	2	3
MSCs+ZnO-L	0	2	2
ZnO-H	0	2	2
ZnO-L	0	0	0

**Table 7 jox-15-00119-t007:** The number of all differentially methylated CpG sites (both unique and common) identified in DNA from untreated wounded skin and in tissue samples after various treatment scenarios. A comparison was made with the untreated wounded skin group as a reference.

Group	Differentially Methylated CpG Sites	Affected Genes (N)
MSCs	0	0
MSCs+Ag-H	819	106
MSCs+Ag-L	0	0
Ag-H	0	0
Ag-L	1480	192
MSCs+ZnO-H	13	2
MSCs+ZnO-L	392	50
ZnO-H	0	0
ZnO-L	0	0

## Data Availability

The genomics data presented in the study are openly available in the public repository of the Institute of Experimental Medicine CAS at the following links: mRNA expression data: https://iemcas-my.sharepoint.com/:f:/g/personal/pavel_rossner_iem_cas_cz/EoJabJSlrYtMlRwU0YV9mLYB5lWMq_LLbiw0oPHrnpuHpQ?e=9lfofu, accessed on 17 July 2025; miRNA expression data: https://iemcas-my.sharepoint.com/:f:/g/personal/pavel_rossner_iem_cas_cz/ElqdNVkwrfNKnHqG087_cE0BNn9xLTEEmojBdDqS6NoBNg?e=OlHySq, accessed on 17 July 2025; DNA methylation data: https://iemcas-my.sharepoint.com/:f:/g/personal/pavel_rossner_iem_cas_cz/EvoJEE9qsMVOkqJgYRBGagoBMoGTOq1bTUb0dDvLF118fA?e=rwtmd6, accessed on 17 July 2025.

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
