# Peer review of "Skin Wound Healing: The Impact of Treatment with Antimicrobial Nanoparticles and Mesenchymal Stem Cells"

_jox, 2025, doi:10.3390/jox15040119_

Round 1

Reviewer 1 Report

Comments and Suggestions for Authors
  1. The introduction provides a complete overview of MSCs in wound healing. Could the authors clarify whether the cited studies refer to preclinical models, clinical trials, or both? Additionally, can the authors briefly comment on how current MSC delivery methods influence therapeutic efficacy.
  2. The authors present an insightful analysis of transcriptomic changes induced by different treatment conditions. It would be helpful to include a brief discussion on the biological relevance of the key deregulated genes mentioned (e.g., Acot11, Lair1, Pyhin1, Sprr2b, Aldh6A1). Specifically, how might these genes contribute to wound healing or inflammatory regulation.
  3. The gene expression data are clearly presented and support the conclusion that Ag NP treatment alone (Ag-L) may promote regenerative responses via increased Col1 expression. However, the authors should clarify whether the observed differences in other genes (e.g., Lgals3, Ccl-2, Mmp-2) reached statistical significance compared to the injured control. Please include the statistical annotations for all comparisons.
  4. The data suggest DNA methylation changes in response to Ag and ZnO NPs, but the study lacks a clear link between methylation changes and gene expression modulation. Authors should identify and discuss specific genes where methylation status correlates with altered expression, especially within relevant healing pathways (e.g., PI3K/AKT, keratinization).
  5. The manuscript describes the exposure of MSCs to Ag and ZnO nanoparticles for 7 days to assess phenotypic changes. However, it is unclear whether this treatment impacts the functional properties of MSCs, such as viability, proliferation, or differentiation potential. Could the authors provide data or comment on whether nanoparticle exposure alters these key MSC functions, beyond surface marker expression
  6. Did the authors validate the efficiency and specificity of each primer pair used in the qRT-PCR assay, such as by generating standard curves

Author Response

1. The introduction provides a complete overview of MSCs in wound healing. Could the authors clarify whether the cited studies refer to preclinical models, clinical trials, or both? Additionally, can the authors briefly comment on how current MSC delivery methods influence therapeutic efficacy.

Response: The referenced studies are mostly reviews (Zeng et al., 2018; Lo et al., 2023; Bellu et al., 2021) that comment on application MSC or NP in wound healing. Generally, MSC and NP separately are used in both preclinical and clinical studies; however, the combined treatment has not been used so far. As for methods of MSC delivery and treatment efficacy, we added a short information to the introduction. The route of dellivery is critical for the success of MSC treatment. Currently, local administration directly to the target organ/tissue or systemic delivery by IP/IV injection is used. The optimal delivery method depends on the specific disease, target tissue and the therapeutic outcome.

2. The authors present an insightful analysis of transcriptomic changes induced by different treatment conditions. It would be helpful to include a brief discussion on the biological relevance of the key deregulated genes mentioned (e.g., Acot11, Lair1, Pyhin1, Sprr2b, Aldh6A1). Specifically, how might these genes contribute to wound healing or inflammatory regulation.

Response: We added a brief information on the key genes to section 4.1.

3. The gene expression data are clearly presented and support the conclusion that Ag NP treatment alone (Ag-L) may promote regenerative responses via increased Col1 expression. However, the authors should clarify whether the observed differences in other genes (e.g., Lgals3, Ccl-2, Mmp-2) reached statistical significance compared to the injured control. Please include the statistical annotations for all comparisons.

Response: No, there was no significant difference in the expression of these genes, with the exception of Col1. The information is now explicitly mentioned in the Figure 6. caption.

4. The data suggest DNA methylation changes in response to Ag and ZnO NPs, but the study lacks a clear link between methylation changes and gene expression modulation. Authors should identify and discuss specific genes where methylation status correlates with altered expression, especially within relevant healing pathways (e.g., PI3K/AKT, keratinization).

Response: We thank the reviewer for this important comment. We did correlation analysis of mRNA expression and DNA methylation as requested and found an association for three genes: pyruvate carboxylase (Pcx), urate (5-hydroxyiso-) hydrolase (Urah) and oestrogen receptor alpha (Esr1). Further literature search showed that there is no causal link between Pcx/Urah expression and wound healing processes. Thus, we did not investigate these genes further. For Esr1, a significant role in wound healing, particularly in the ability to promote alternative macrophage activation, which is beneficial for timely healing, was reported . When mRNA expression and DNA methylation data were checked, we found decreased CpG methylation (5 CpG sites) and downregulation od Esr1 mRNA. This is not an expected result, as decreased CpG methylation should be associated with increased mRNA expression. However, there reason might be that other CpG sites in the Esr1 gene, not detected here, and/or other regulatory mechanisms were involved. Thus, we conclude that no direct correlation between deregulated mRNA and CpG methylation can be detected in our study. We added this information to the manuscript (section 3.2.).

5. The manuscript describes the exposure of MSCs to Ag and ZnO nanoparticles for 7 days to assess phenotypic changes. However, it is unclear whether this treatment impacts the functional properties of MSCs, such as viability, proliferation, or differentiation potential. Could the authors provide data or comment on whether nanoparticle exposure alters these key MSC functions, beyond surface marker expression

Response: Yes, there were effects of NP on functional properties of MSCs. Before conducting the in vivo study, we performed an in vitro assessment of potential effects of NP on MSCs and found that metal NP inhibit therapeutic properties of MSCs by a direct negative effect on their secretory activity, but MSCs have preserved the ability to stimulate cytokine and growth factor production by macrophages [1,2]. In the first paragraph of Discussion we mention that NP for skin injury experiments were selected based on a previous study in which the effects of metal NPs on the characteristics and function properties of MSCs were analysed; the study is referenced.

6. Did the authors validate the efficiency and specificity of each primer pair used in the qRT-PCR assay, such as by generating standard curves.

Response: Yes, we do that as a part of our standard experimental procedures.

References:

  1. Echalar, B.; Dostalova, D.; Palacka, K.; Javorkova, E.; Hermankova, B.; Cervena, T.; Zajicova, A.; Holan, V.; Rossner, P. Effects of Antimicrobial Metal Nanoparticles on Characteristics and Function Properties of Mouse Mesenchymal Stem Cells. Toxicol. In Vitro 2023, 87, 105536, doi:10.1016/j.tiv.2022.105536.
  2. Holan, V.; Cervena, T.; Zajicova, A.; Hermankova, B.; Echalar, B.; Palacka, K.; Rossner, P.; Javorkova, E. The Impact of Metal Nanoparticles on the Immunoregulatory and Therapeutic Properties of Mesenchymal Stem Cells. Stem Cell Rev. Rep. 2023, doi:10.1007/s12015-022-10500-2.

Reviewer 2 Report

Comments and Suggestions for Authors

See reviewer comments in the file.

Author Response

This manuscript examines the effects of nanoparticles on wound healing using an in vitro study with tissue extracted from mouse wounds. It is an interesting and detailed in vitro study. Although this study presents a lot of useful information, reviewer would like to refer to a few points that are unclear.

Queries:

  1. The author created the wound by removing tissue from the back of the mouse. Since the subcutaneous tissue of the mouse back is extremely thin, most of the tissue removed is considered to be epidermis and dermis. This is called a shallow wound. In wound healing, the histological morphology of the regenerated tissue depends on the depth and size of the wound. The author should define the depth or wound cross-section of the created wound.

Response: We greatly appreciate the valuable feedback regarding the depth and size of wound. In all cases, the wound involved the epidermis, dermis and part of hypodermis with majority of fat tissue. The size of injury was approximately 6 mm in diameter and the average thickness of excised skin was about 540 µm. This information was added to the manuscript (Methods, section 2.2).

  1. Figure 1 in the Supplement shows that it basically corresponds to dry healing. In this case, have the author observed any scab formation? If authors have confirmed the formation of a scab, the tissue will regenerate just below the scab. Is it safe to assume that the scab is not included in the collected tissue? Because the author states that Ag was associated with desmosome organization. The desmosome organization is associated with the acab formation.

Response: We thank the Reviewer for this important comment. The bandage of wound contained paraffin gauze dressing which is associated with moist wound healing. We added this information into manuscript (Methods, section 2.2).

During our experiments, we did not observe an apparent formation of solid scabs because all wounds were covered with paraffin bandages to protect them from drying and keeping their wetness. Thus, the collected tissue for analysis did not contain solid scab. Even though some desmosome organization was detected, the level of scab formation was not macroscopically observed.

  1. The author conducted ten different tests on wound healing. In Supplementary Figure 3, the author needs to show photos of all the mice's skin on day 7 postoperatively. Is there good quality granulation seen in the proliferative phase or is it poor white granulation? Reviewer wanted to know what kind of granulation was taken.

Response: Thank you for this important comment. Unfortunately, we are not able to supplement additional images of wounds after 7 days for 10 different types of treatments. Since all wounds looked macroscopically similar, we took picture only of some representative wounds. The tissue from wound was collected already on day 7 after the injury during progression from the inflammatory to proliferative phase of healing, the formation of granulation tissue was in the initial phase and we were not able to determine the type of granulation (it was not good quality granulation which is typical for later proliferative phase).

  1. On Page of L648, the author concluded that the ZnO nanoparticles were partially dissolved and that the Ag remained in the particle form. ZnO is insoluble in the body pH levels. On the other hand, Ag is partially ionized at body pH levels. Supporting data must be provided.

Response: This statement was a speculation by which we tried to explain the different results for ZnO and Ag NP. We did not look further to the fate of NP in the organism as it was out of the scope of the study. We modified the text in the manuscript (Conclusions) to make these points clear.

Comment: The concern as reviewer is whether the tissue taken from the injury site follows the proper wound healing process. In wound healing with these materials, it is itself a foreign substance, and its introduction into regenerating tissue may lead to not only delayed healing but also chronic inflammation. And also, the antimicrobial activity of ZnO is related to the concentration of reactive oxygen species generated from its surface; that of Ag is attributed to the Ag ions leached from its surface. Nanoparticles can be taken up into cells by endocytosis and have a significant impact on intracellular organs. Angiogenesis in regenerating tissue is suppressed as the concentration of Ag ions increases. Without a close examination of these details, a detailed verification may not be possible. Although this study assumes that the disease is in the inflammatory to proliferative phase, the reviewer would like to see verification of this phase based on stained images of tissue sections. Since there are many pathological conditions and forms of wounds, we look forward to future research.

Response: We thank the Reviewer for this comment and agree that stained images of the tissue sections would give definite answer on the healing progress. However, these steps were beyond the intended scope of our research, as we were primarily looking at molecular/cellular changes following MSC/NP application regardless identification the healing progress. We believe that according to the results from flow cytometry, ELISA and real-time PCR we can assume that inflammatory phase was ending (minor increase of IL-6 or some leukocyte populations) and proliferative phase was starting to prevail (collagen and growth factors expression levels). The Reviewer’s suggestions are valuable inputs for future research.
